# The Change in Social Eating over Time in People with Head and Neck Cancer Treated with Primary (Chemo)Radiotherapy: The Role of Swallowing, Oral Function, and Nutritional Status

**DOI:** 10.3390/cancers15051603

**Published:** 2023-03-04

**Authors:** Aurora Ninfa, Femke Jansen, Antonella Delle Fave, Birgit I. Lissenberg-Witte, Nicole Pizzorni, Robert J. Baatenburg de Jong, Femke Lamers, C. René Leemans, Robert P. Takes, Christianus H. J. Terhaard, Antonio Schindler, Irma M. Verdonck-de Leeuw

**Affiliations:** 1Department of Pathophysiology and Transplantation, University of Milan, 20122 Milan, Italy; 2Department of Otolaryngology-Head and Neck Surgery, Amsterdam UMC Location Vrije Universiteit, 1081 HV Amsterdam, The Netherlands; 3Amsterdam Public Health Research Institute, Mental Health, 1081 HV Amsterdam, The Netherlands; 4Cancer Center Amsterdam, Treatment and Quality of Life, 1081 HV Amsterdam, The Netherlands; 5Department of Epidemiology and Data Science, Amsterdam UMC Location Vrije Universiteit, Boelelaan 1117, 1081 HV Amsterdam, The Netherlands; 6Department of Biomedical and Clinical Sciences, University of Milan, 20157 Milan, Italy; 7Department of Otorhinolaryngology, Erasmus MC, University Medical Center, 3015 GD Rotterdam, The Netherlands; 8Department of Psychiatry, Amsterdam UMC Location Vrije Universiteit Amsterdam, Boelelaan 1117, 1081 HV Amsterdam, The Netherlands; 9Department of Otorhinolaryngology-Head and Neck Surgery, Radboud University Medical Center, 6525 GA Nijmegen, The Netherlands; 10Department of Radiation Oncology, UMC Utrecht, 3584 CX Utrecht, The Netherlands; 11Department Clinical, Neuro and Developmental Psychology, Vrije Universiteit Amsterdam, Van der Boechorststraat 7-9, 1081 HV Amsterdam, The Netherlands

**Keywords:** head and neck cancer, (chemo)radiotherapy, social eating, swallowing, nutritional status

## Abstract

**Simple Summary:**

Social eating problems may affect lives of people with head and neck cancer (HNC) during and after (chemo)radiotherapy treatment. This study aimed at investigating if people with HNC experience social eating problems from diagnosis up to 24 months after (chemo)radiotherapy and if their change over time is associated with swallowing, oral function, and nutritional status, in addition to the clinical, personal, physical, psychological, social, and lifestyle dimensions. We found that social eating problems worsened three months after treatment and improved to baseline levels up to 24 months. The change in social eating problems over time was associated with swallowing, nutritional status, tumor subsite, age, muscle strength, hearing problems, and depressive symptoms. Results are relevant to research and clinical practice for improving personalized supportive care targeting social eating problems.

**Abstract:**

This study aimed at investigating the change in social eating problems from diagnosis to 24 months after primary (chemo)radiotherapy and its associations with swallowing, oral function, and nutritional status, in addition to the clinical, personal, physical, psychological, social, and lifestyle dimensions. Adult patients from the NETherlands QUality of life and BIomedical Cohort (NET-QUBIC) treated with curative intent with primary (chemo)radiotherapy for newly-diagnosed HNC and who provided baseline social eating data were included. Social eating problems were measured at baseline and at 3-, 6-, 12-, and 24-month follow-up, with hypothesized associated variables at baseline and at 6 months. Associations were analyzed through linear mixed models. Included patients were 361 (male: 281 (77.8%), age: mean = 63.3, SD = 8.6). Social eating problems increased at the 3-month follow-up and decreased up to 24 months (F = 33.134, *p* < 0.001). The baseline-to-24 month change in social eating problems was associated with baseline swallowing-related quality of life (F = 9.906, *p* < 0.001) and symptoms (F = 4.173, *p* = 0.002), nutritional status (F = 4.692, *p* = 0.001), tumor site (F = 2.724, *p* = 0.001), age (F = 3.627, *p* = 0.006), and depressive symptoms (F = 5.914, *p* < 0.001). The 6–24-month change in social eating problems was associated with a 6-month nutritional status (F = 6.089, *p* = 0.002), age (F = 5.727, *p* = 0.004), muscle strength (F = 5.218, *p* = 0.006), and hearing problems (F = 5.155, *p* = 0.006). Results suggest monitoring social eating problems until 12-month follow-up and basing interventions on patients’ features.

## 1. Introduction

Head and neck cancer (HNC) has a worldwide incidence of 798,577 new cases/year and a high mortality rate (500,000 deaths/year) [1]. Treatment of HNC is associated with functional loss of the swallowing organs and often leads to swallowing and oral dysfunctions as well as poor nutritional status [2,3,4,5,6].

Compared to the general population and people undergoing other treatments for HNC [7,8], patients treated with (chemo)radiotherapy ((C)RT) experience lower health-related quality of life (HRQOL) [9,10], associated with social eating problems [11]. Social eating, also known as commensality (i.e., coming together at a table and eating and drinking with other people), is a human experience that usually takes place daily and as part of celebratory occasions [12]. Social eating problems include difficulties with meal consumption and restricted participation at mealtime. Social eating problems among people with HNC treated with (C)RT are extensively reported in experimental studies [13,14] and literature reviews [15,16]. A cross-sectional and prospective study found that these problems are directly associated with swallowing dysfunctions [17] and poor nutritional status up to one year after treatment [18], respectively. Additional studies are, however, needed to comprehensively identify the variables associated with the change in social eating problems over time by measuring them at relevant time-points of the cancer care. Social eating is a multifaceted construct [19], thus we hypothesize that clinical, personal, physical, psychological, social, and lifestyle dimensions at the start of and during (C)RT may be associated with its change over time.

This study aimed to investigate whether (i) people with HNC treated with primary (chemo)radiotherapy experience changes in social eating problems from diagnosis up to 24 months after treatment; (ii) the change in social eating problems from baseline to 24-month follow-up is associated with patient conditions at the beginning of the cancer treatment trajectory with respect to swallowing-related quality of life and symptoms, oral function, nutritional status (i.e., HNC-specific), clinical, personal, physical, psychological, social, and lifestyle variables; and (iii) the change in social eating problems from 6 to 24 months after treatment is associated with treatment consequences with respect to the following: swallowing-related quality of life and symptoms, oral function, nutritional status (i.e., HNC-specific), as well as clinical, personal, physical, psychological, social, and lifestyle variables, as assessed at 6 months after treatment. The results of this study are relevant to improving supportive care targeting social eating problems among people with HNC.

## 2. Materials and Methods

### 2.1. Participants and Procedures

Data were used from the NETherlands QUality of life and BIomedical Cohort (NET-QUBIC) study, which longitudinally collected an extensive set of biosamples, fieldwork data (interviews, functional tests), and patient-reported outcome measures [20,21]. Participants were recruited with consecutive sampling in Dutch HNC centers between 2014 and 2018, with up to 5-year follow-ups. Inclusion criteria were (i) newly diagnosed HNC (oral cavity, oropharynx, hypopharynx, larynx, and unknown primary, at all stages); (ii) treatment with curative intent (all modalities); (iii) age > 18 years, (iv) ability to write, read, and speak Dutch. Exclusion criteria were: (i) lymphoma, skin malignancies, thyroid cancer; (ii) inability to understand the questions or test instructions; and (iii) severe psychiatric comorbidities. Participants were included in this specific study when they (i) underwent primary (chemo)radiotherapy and (ii) completed the Social Eating Scale of the European Organization for Research and Treatment of Cancer Quality of Life Questionnaire head and neck cancer-specific module (EORTC QLQ-H&N35) at baseline (before treatment). The study protocol was approved by the ethical committee of VUmc (2013.301(A2018.307)-NL45051.029.13). All participants signed a form of informed consent. NET-QUBIC patients were treated according to the current standard as defined by Dutch HNC guidelines on diagnosis, treatment, and follow-up care.

In the present study, data were used on social eating (baseline, 3-, 6-, 12-, and 24-month follow-up), HNC-specific physical, psychological, social, and lifestyle (dynamic) variables (baseline and 6-month follow-up), and personal, clinical, and psychological (static) variables (baseline).

### 2.2. Outcome Measures

Detailed information and references of all collected outcomes are available in the NET-QUBIC data catalogue (https://researchers.kubusproject.nl/data-catalogue accessed on 20 September 2022).

The primary outcome was the Social Eating Scale of the EORTC-QLQ-HN35 [22]. It consists of four items describing troubles when eating with family members or other people and meal enjoyment, with a scale score range of 0–100 and higher scores indicating more problems.

The HNC-specific variables were measured via the Swallowing Quality of Life Questionnaire (SWAL-QoL) [23] (swallowing-related QoL total score, and Symptoms subscale score, respectively), the Functional Rehabilitation Outcome Grades (FROG) [24] (oral function, total score), and the Mini Nutritional Assessment (MNA) [25] (nutritional status, total score). The SWAL-QoL (23 items) assesses problems with preparing and consuming meals, emotional response, and restriction in social functioning related to swallowing, whereas the SWAL-QoL Symptoms subscale (14 items) assesses oral and pharyngeal swallowing symptoms. Scale scores range from 0 to 100, higher scores indicating lower QoL and more symptoms. The FROG comprises a clinical assessment of the mandible, teeth, lips, tongue, oropharynx, shoulder, and xerostomia; scores range from 0 to 80, higher scores indicating better function. The MNA categorizes patients as either normally nourished (score > 24) or malnourished or at risk of malnutrition (≤23.5).

Clinical variables included tumor site (oropharynx/hypopharynx/larynx/oral or unknown primary), tumor stage (0-I/II/III/IV), HPV status (positive/negative), chemotherapy treatment (yes/no), comorbidity (Adult Comorbidity Evaluation-27 Index, none-mild/moderate-severe), major depression disorder in lifetime (yes/no), and WHO performance status (normal (score = 0)/restricted (score > 0)).

Personal variables included age, sex (self-reported, male/female), education (low/middle/high), living arrangements (living together/alone), and personality (NEO Five Factor Inventory; higher scores indicate higher levels of neuroticism, extraversion, openness to experience, agreeableness, or conscientiousness).

Physical variables were muscle strength (handgrip test, </≥ 25th percentile adjusted for age and sex), pulmonary function (maximum peak expiratory flow), independence in instrumental activities of daily living (IADL; higher scores indicating higher independence), perceived hearing problems (Caron questionnaire), and physical fatigue (Multidimensional Fatigue Inventory (MFI); higher scores indicating more problems and fatigue).

Psychological variables were distinguished into resources and problems. Resources included measures of positive adjustment to cancer (mental adjustment to cancer; positive summary score), general coping strategies of active coping, palliative coping, seeking support, and comforting thoughts (Utrecht Coping List (UCL)), personal control (Pearlin and Schooler’s Sense of Mastery Scale), and general self-efficacy (GSE), with higher scores indicating stronger positive adjustment and adaptive coping strategies and higher mastery and self-efficacy. Psychological problems included cognitive failure (the Cognitive Failure Questionnaire), sleep problems (the Pittsburgh Sleep Quality Index), mental fatigue (MFI), depression and anxiety symptoms (the Hospital Anxiety and Depression Scale), and fear of cancer recurrence (the Cancer Worry Scale), with higher scores indicating more problems and symptoms. Moreover, negative adjustment to cancer (MAC; negative summary scores) and general coping strategies of avoidance, passive coping, and expression of negative emotions (UCL) were measured, with higher scores indicating stronger negative adjustment and the use of maladaptive coping strategies.

Social variables included social support (Social Support List-Interactions, with higher scores indicating higher support), loneliness (the Loneliness Scale (deJong-Gierveld) with higher scores meaning higher loneliness), problems with social contacts (EORTC QLQ-HN35), and financial problems (EORTC QLQ C30) with higher scores indicating more problems.

Lifestyle variables were current daily smoking (yes/no), excessive alcohol consumption (yes/no, women/men: >14/ > 21 glasses/week), body mass index (BMI), physical activity (Physical Activity Scale for the Elderly, with higher scores indicating more frequent physical activity), and stress (Impact of Event Scale-Revised, with higher scores indicating more intrusion, avoidance, hyperarousal, numbing, and sleep disturbance).

### 2.3. Statistical Analyses

Participants’ baseline characteristics were compared to those of NET-QUBIC patients treated with (C)RT without social eating data (independent samples *t*-test for normally distributed continuous variables, Mann-Whitney U test for non-normally distributed variables, chi-square test for categorical variables). *p*-values <0.05 were considered statistically significant. Descriptive statistics were generated for all included outcomes.

Linear mixed models (LMM) with fixed effects for time (categorical), random intercept for subjects, no random slope, and a 99% confidence interval (CI) (compound symmetry matrix) were used to investigate the change in social eating problems from baseline through 3, 6, 12, and 24 months after treatment. *p*-values <0.01 were considered statistically significant.

Associations were assessed (i) between the change in social eating from baseline to 24 months after treatment and variables measured at baseline and (ii) between the change in social eating from 6 to 24 months and baseline static and 6-month dynamic variables. LMM with fixed effects for time, the potential factor(s), their two-way interaction time*factor, a random intercept for subjects, no random slope, and 95% CI were computed. Significant two-way interactions (*p* < 0.01) indicated an association between the factor(s) and the change in social eating problems over time. To investigate associations among combinations of factors, within-domain and across-domain multivariable models were computed using a backward selection procedure. An interaction time*factor with *p* <0.05 was set as the threshold for factors to be entered and retained in the model, whereas, to handle multiple comparisons, associations with the change in social eating problems over time were interpreted as significant when *p* < 0.01.

The direction and time-point(s) of the associations were illustrated by plotting the estimated course of social eating, with respect to each significant factor. For continuous variables, 25–50–75th percentiles were plotted as exemplary low, moderate, and high values.

Statistical analyses were conducted using IBM Statistical Package for the Social Science (SPSS) version 26.0 (IBM Corporation, Armonk, NY, USA). Figures were created using SPSS, Microsoft Excel (Microsoft Corporation, Redmond, WA, USA, 2018), and the R package “dagitty” [26].

## 3. Results

### 3.1. Participants

Of the 739 participants included in the NET-QUBIC study, 457 were treated with (C)RT. Among them, 361 (79.0%) completed the EORTC questions on social eating at baseline and were included in this study. Included patients more often cohabited, had a lower tumor stage, and fewer comorbidities compared to excluded patients (*p* < 0.05). Table 1 offers an overview of the study population’s features in the clinical, personal, physical, psychological, lifestyle, and social domains at baseline (all variables) and at 6 months after treatment (dynamic variables only). The flow diagram of the study, including reasons for dropout is provided in Figure 1.

### 3.2. The Change in Social Eating Problems over Time

Patients experienced significant changes in social eating problems from baseline to 24 months after treatment (time: F = 33.134, *p* < 0.001). The mean social eating score amounted to 10.8 (SD = 15.6, min-max 0–91.7) before treatment, increased (worsened) at three months after treatment, slowly decreased (improved) from 6 to 12 months after treatment, and returned to baseline level at 24 months after treatment. Figure 2 illustrates the change in social eating problems over time, while observed and estimated social eating scores are provided in Table 2.

### 3.3. Associations with Swallowing-Related QoL and Symptoms, Oral Function, and Nutritional Status

Results of univariable and multivariable within-domain LMM analyses regarding the association between change in social eating problems over time and swallowing-related QoL, swallowing symptoms, oral function, and nutritional status measured at baseline and at 6 months are presented in Table 3. In the univariable and multivariable analyses, only swallowing-related QoL was significantly inversely associated with the change in social eating problems from baseline to 24 months after treatment (*p* < 0.001). Conversely, none of the investigated variables was significantly associated with changes in social eating from 6 to 24 months after treatment (*p* < 0.01) in either the univariable or multivariable analyses.

### 3.4. Associations Within- and Across-Domains

Results of the univariable and multivariable LMM analyses within each domain and of the multivariable LMM analyses across-domains are presented in Table 3. Figure 3 and Figure 4 illustrate the change in social eating problems over time in relation to the significantly associated variables (*p* < 0.01) in the multivariable across-domains analyses at baseline and 6 months after treatment, respectively.

#### 3.4.1. The Change in Social Eating Problems from Baseline to 24 Months after Treatment

Results of the within-domain multivariable analyses showed that clinical (tumor location), personal (age and the personality trait of neuroticism), and psychological (depressive symptoms) variables were significantly associated with the change in social eating problems over time (*p* < 0.01).

In the overall model, all domains were represented by at least one variable (*p* < 0.05). After the backward selection procedure, swallowing-related QoL, swallowing symptoms, nutritional status, tumor location, age, and depressive symptoms at baseline were significantly associated with the change in social eating problems over time (*p* < 0.01).

As shown in Figure 3, five different patterns emerged: (a) more social eating problems at all time-points were associated with being treated for oropharyngeal, oral, and unknown primary tumors compared to laryngeal tumors; (b) increased (worse) social eating problems from baseline to 3 months after treatment were associated with lower swallowing-related QoL, whereas patients with higher swallowing-related QoL at baseline reported more social eating problems at 12 months; (c) a steeper increase(worsening) in social eating problems at 3 months after treatment was associated with a normal nutritional status, while more social eating problems from 6 to 24 months were associated with a poor nutritional status; (d) a steeper increase (worsening) in social eating problems at 3 months was associated with more depressive symptoms at baseline, followed by a realignment to the course characterizing patients with less depressive symptoms; and (e) after increasing (worsening) social eating problems at 3 months, a flatter decrease (slower improvement) from 6 to 24 months was associated with more swallowing symptoms and older age at baseline.

#### 3.4.2. The Change in Social Eating Problems from 6 to 24 Months after Treatment

Results of the within-domain multivariable analyses showed that the personal (age), physical (perceived hearing problems), psychological (fear of cancer recurrence), and lifestyle (stress symptoms of sleep disturbance) domains were associated with the change in social eating problems from 6 to 24 months after treatment (*p* ≤ 0.01).

In the overall model, all domains except for the social one were represented by at least one variable (*p* < 0.05). After the backward selection procedure, nutritional status, age, muscle strength, and hearing problems were significantly associated with the change in social eating problems over time (*p* < 0.01). As shown in Figure 4, three different patterns emerged: (a) increased (worse) social eating problems at 24 months after treatment compared to 6 months were associated with higher age and reduced muscle strength; (b) a steeper decrease (improvement) in social eating problems at 24 months was associated with a poor nutritional status, despite the fact that these patients maintained higher social eating problems over time compared to normally nourished patients; (c) a steeper decrease (improvement) in social eating problems at 12 months after treatment was associated with having more hearing problems, followed by a realignment to the course characterizing patients with less hearing problems.

## 4. Discussion

This study showed that among HNC survivors treated with (C)RT, social eating problems significantly increased at 3 months after treatment and decreased (improved) to baseline levels up to 24 months, in accordance with previous evidence [9,27]. The 24-month change in social eating problems was significantly associated with HNC-specific (swallowing-related QoL and symptoms, and nutritional status), clinical (tumor subsite), personal (age), physical (muscle strength and hearing problems), and psychological features (depressive symptoms). Based on these results causal relationships with change in social eating problems may be hypothesized for variables as measured at baseline and 6 months after treatment, as shown in Figure 5. Previous evidence detected associations between the 12-month change in social eating problems and tumor site, age, sex, cohabitation, and socio-economic status [27]. Discrepancies may be related to the longer observation period, the focus on (C)RT, and the broader set of variables considered in our study.

Major clinical implications can be derived from this study to improve social eating supportive care.

### 4.1. Regular Monitoring over Time Matters

Results suggest that social eating should be monitored at least throughout the first year of the cancer care trajectory and possibly also beyond this period. People with HNC may continue to suffer from social eating problems compared to the general population [9], due to late treatment toxicities affecting swallowing function [2,28].

### 4.2. Advocating a Personalized Approach

A complex picture emerged from this study, suggesting patients with specific features at relevant time-points (baseline and 6-month follow-up) may be more vulnerable to social eating problems. Variables with documented cross-sectional associations with social eating [17,18] (i.e., swallowing-related QoL and symptoms and nutritional status) were related to its change over time. Associations were also detected for clinical (tumor site), personal (age), physical (muscle strength and hearing problems), and psychological (depressive symptoms) variables. Over time, association patterns with social eating problems differed across variables. Associations were observed between more depressive and swallowing symptoms at baseline and more (worse) social eating problems in the early (3 months) and later phases (6–24 months). Considering that in similar HNC cohorts, high depressive and swallowing symptoms were described for up to 3- and 6-month follow-up [29,30], it may be hypothesized that worse social eating problems are observed as long as these symptoms persist. Older age was associated with increasing (worsening) social eating problems from 12- to 24-month follow-up. Patterson et al. described an opposite trend, their investigation being limited to the first year after diagnosis and including patients treated with all modalities [27]. Our findings support prolonged monitoring of social eating in elderly patients. Consistently with previous evidence [27], patients with laryngeal tumors reported fewer social eating problems over time compared to patients with oropharyngeal tumors. Possible explanations are the functional damage to oropharyngeal organs associated with (C)RT [3] and the longer eating duration reported by patients with oropharyngeal cancer [30]. Some ambivalent associations emerged regarding swallowing-related QoL and nutritional status, suggesting that these measures may not allow for clearly identifying patients eligible for social eating supportive care. Consistently with previous evidence on reduced swallowing-related QoL at 3-month follow-up [30] and on associations between malnutrition and worse social eating problems up to 1-year follow-up [18], we found associations between worse social eating problems and low swallowing-related QoL at baseline and poor nutritional status at baseline and 6 months. Similarly, patients with high swallowing-related QoL and normal nutritional status at baseline presented increased (worse) social eating problems at different points in their cancer care, which we hypothesized was due to (C)RT side effects. Finally, we found a steeper decrease (faster improvement) in social eating problems up to 12 months for patients with more hearing problems at 6-month follow-up, whereas those with reduced muscle strength at 6 months experienced worse social eating problems at 12-month follow-up. These associations require further exploration and should be interpreted with caution due to the lack of previous evidence.

In summary, timely identifying people with HNC at risk of social eating problems is a challenging task, involving several biomedical and psychosocial dimensions.

### 4.3. Limitations and Future Directions

Due to the small number of patients (N < 15), those who had or were at risk of malnutrition and had oral or unknown primary tumors were aggregated into a single category, neglecting potential differences in social eating patterns and thus limiting the interpretation of related findings. Furthermore, included NET-QUBIC patients had less severe tumor stages and comorbidity and more often cohabited with other people compared to excluded patients, raising representativeness issues.

Studies targeting different cohorts with balanced characteristics are needed to confirm our findings and foster their external validity. Future studies should identify variables associated with clinically relevant social eating problems up to and beyond 24-month follow-up and characterize patients at risk of persistent and increased social eating problems.

## 5. Conclusions

This study provides evidence of the multifaceted nature of social eating difficulties up to 24 months after primary (C)RT treatment for HNC, highlighting their changes over time and their association with HNC-specific (swallowing problems and nutritional status), clinical (tumor subsite), personal (age), physical (muscle strength and hearing problems), and psychological features (depressive symptoms). Results suggest social eating should be targeted on average until 12 months after treatment. In addition to the symptomatic patients, those with high swallowing-related QoL, normal nutritional status, and an oropharyngeal tumor at baseline, as well as those with an older age and reduced muscle strength at 6 months, should be closely monitored up to 24 months. The course of social eating beyond 24 months after (C)RT treatment and associated variables is to be explored.

## Figures and Tables

**Figure 1 cancers-15-01603-f001:**
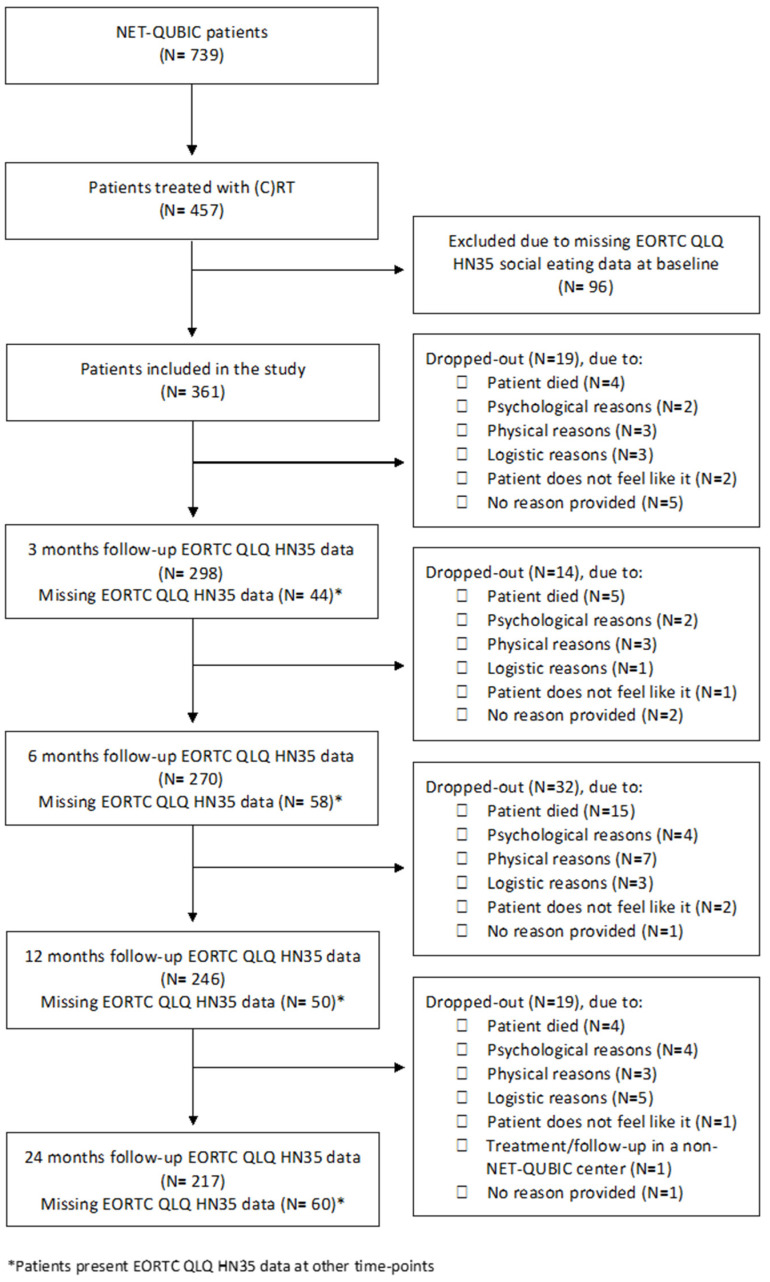
The study flow diagram, with reasons for exclusion at each time point.

**Figure 2 cancers-15-01603-f002:**
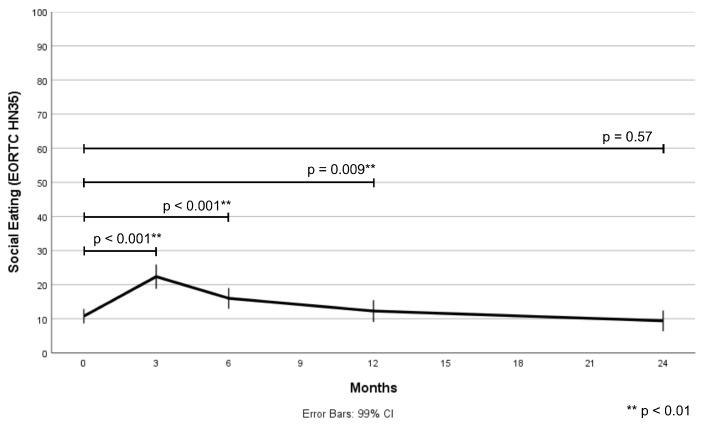
The change in social eating problems (EORTC QLQ HN35) from baseline to 24 months after treatment.

**Figure 3 cancers-15-01603-f003:**
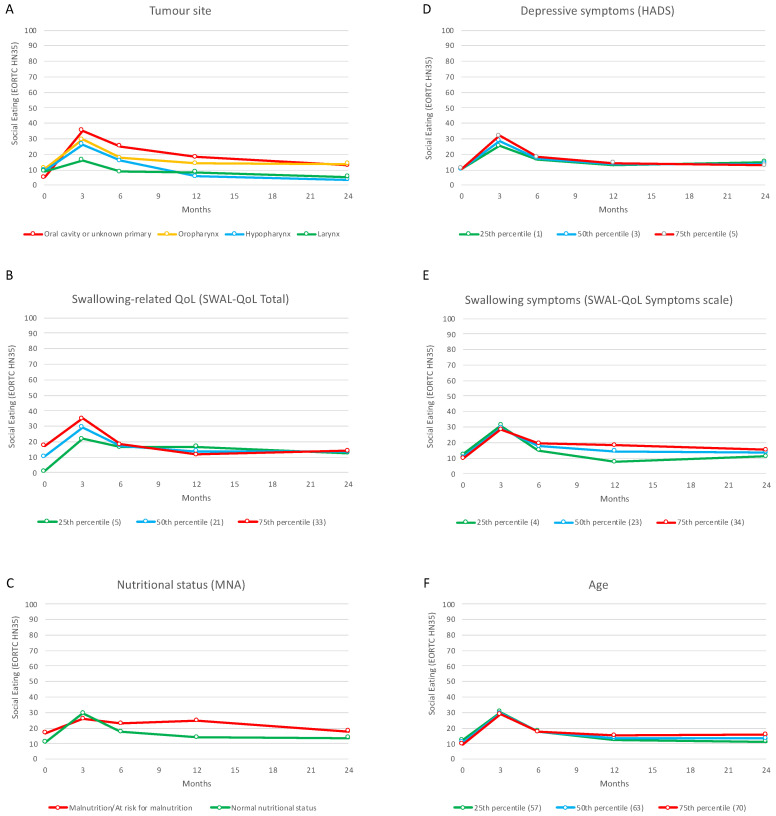
The change in social eating problems (EORTC QLQ HN35) from baseline to 24 months after treatment in relation to tumor site (**A**), swallowing-related Quality of life (**B**), nutritional status (**C**), depressive symptoms (**D**), swallowing symptoms (**E**), and age (**F**) at baseline.

**Figure 4 cancers-15-01603-f004:**
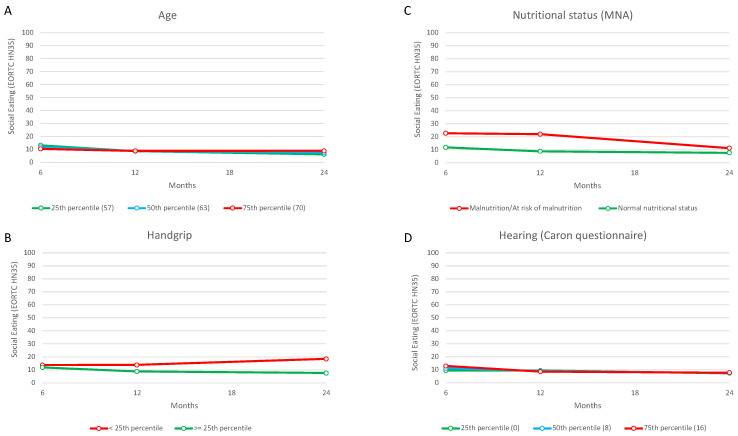
The change in social eating problems (EORTC QLQ HN35) from 6 to 24 months after treatment in relation to age (**A**), muscle strength (**B**), nutritional status (**C**), and hearing problems (**D**) at 6 months.

**Figure 5 cancers-15-01603-f005:**
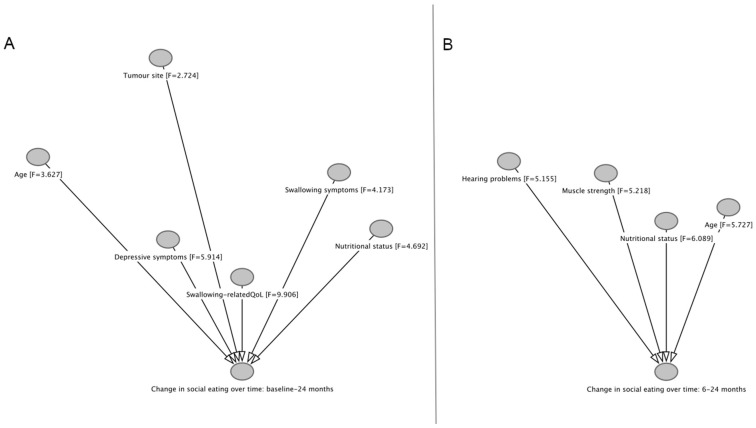
Patients’ features with hypothesized causal relationships with changes in social eating problems from baseline to 24 months (**A**) and from 6 to 24 months follow-up (**B**). Shorter arrows indicate stronger associations.

**Table 1 cancers-15-01603-t001:** Participants’ features at baseline and 6 months post treatment.

N = 361		Baseline		6 Months
	N	% or Mean ± SD (Min-Max)	N	% or Mean ± SD (Min-Max)
HNC-SPECIFIC DOMAIN				
Swallowing-related Quality of Life (SWAL-QoL Total)	342	21.2 ± 17.3 (0–77)	275	23.6 ± 18.1 (0–73)
Perceived swallowing symptoms (SWAL-QoL Symptoms)	339	21.84 ± 17.83 (0–77)	272	26.60 ± 18.95 (0–91)
Oral Function (FROG)	290	72.8 ± 5.7 (44–80)	284	71.0 ± 6.0 (32–80)
Nutritional status (MNA)				
Normal nutritional status	202	56.0%	207	57.3%
At risk of malnutrition or malnourished	87	24.1%	83	23.0%
Missing	72	19.9%	71	19.7%
CLINICAL DOMAIN				
Tumor site				
Oropharynx	185	51.2%	-	-
Hypopharynx	34	9.4%	-	-
Larynx	123	34.1%	-	-
Oral cavity or unknown primary	19	5.3%	-	-
Tumor stage				
0-I	44	12.1%	-	-
II	71	19.7%	-	-
III	71	19.7%	-	-
IV	175	48.5%	-	-
Chemotherapy				
Yes	164	45.4%	-	-
No	197	54.6%	-	-
HPV status				
HPV+	101	28.0%	-	-
HPV-	98	27.1%	-	-
Missing	162	44.9%	-	-
Comorbidity				
None-mild	242	67.0%	-	-
Moderate-severe	106	29.4%	-	-
Missing	13	3.6%	-	-
Performance status (WHO)				
Normal	250	69.3%	-	-
Restricted in physical activity	111	30.7%	-	-
Lifetime depression diagnosis				
Yes	42	11.6%	-	-
No	250	69.3%	-	-
Missing	69	19.1%	-	-
PERSONAL FACTORS				
Age (years)	361	63.3 ± 8.6 (37–86)	-	-
Sex				
Male	281	77.8%	-	-
Female	80	26.9%	-	-
Education level				
Low	140	38.8%	-	-
Middle	91	25.2%	-	-
High	97	26.9%	-	-
Missing	33	9.1%	-	-
Living arrangements				
Living together	244	67.6%	-	-
Living alone	84	23.3%	-	-
Missing	33	9.1%	-	-
Personality (NEO-FFI)				
Neuroticism	348	27.1 ± 7.2 (12–50)	-	-
Extraversion	347	40.3 ± 6.6 (22–60)	-	-
Openness to experience	346	36.0 ± 6.0 (22–58)	-	-
Agreeableness	348	44.2 ± 5.0 (31–58)	-	-
Conscientiousness	348	45.4 ± 5.9 (26–60)	-	-
PHYSICAL DOMAIN				
Muscle strenght (hand grip test)				
<25th percentile	45	12.5%	72	19.9%
>25th percentile	246	68.1%	222	61.5%
Missing	70	19.4%	67	18.6%
Hearing (Caron questionnaire)	338	10.4 ± 10.5 (0–54)	271	11.7 ± 11.7 (0–53)
Physical fatigue (MFI)	340	10.1 ± 4.7 (4–20)	273	10.8 ± 4.7 (4–20)
Independence in daily activities (IADL)	294	7.2 ±1.0 (3–8)	297	7.2 ± 1.0 (3–8)
Peak Expiratory Flow (l/min)	291	409.5 ±145.7 (60–800)	288	400.0 ±142.7 (60–800)
PSYCHOLOGICAL DOMAIN–RESOURCES				
Positive adjustment to cancer (MAC)	343	50.1 ± 6.0 (23–67)	272	49.3 ± 6.7 (24–65)
Coping (UCL) ^1^				
Active coping	348	11.6 ± 3.6 (0–21)	-	-
Palliative coping	348	9.2 ± 3.4 (0–17)	-	-
Seeking support	348	6.8 ± 3.1 (0–17)	-	-
Reassuring thoughts	347	7.2 ± 2.4 (0–15)	-	-
Personal control/Mastery (PSMS) ^1^	347	17.9 ± 3.4 (8–25)	-	-
General Self-efficacy (GSE) ^1^	339	32.8 ± 5.0 (13–40)	-	-
PSYCHOLOGICAL DOMAIN–PROBLEMS				
Cognitive failure (CFQ)	342	23.8 ± 12.6 (0–57)	276	23.3 ± 16.6 (0–75)
Mental fatigue (MFI)	345	8.8 ± 3.8 (4–20)	276	8.6 ± 4.0 (4–20)
Anxiety symptoms (HADS)	359	5.3 ± 3.8 (0–20)	266	3.5 ± 3.5 (0–17)
Depressive symptoms (HADS)	360	3.5 ± 3.4 (0–19)	270	3.2 ± 3.2 (0–13)
Fear of Cancer Recurrence (CWS)	337	13.5 ± 4.3 (8–32)	276	12.6 ± 4.4 (8–31)
Negative adjustment to cancer (MAC)	344	31.9 ± 6.4 (16–60)	273	28.9 ± 7.1 (16–51)
Sleep problems (PSQI)	338	5.5 ± 3.6 (0–18)	274	5.1 ± 3.8 (0–19)
Coping (UCL) ^1^				
Avoidance coping	347	7.3 ± 3.2 (0–17)	-	-
Passive coping	348	3.2 ± 2.7 (0–13)	-	-
Expression of negative emotions	344	1.7 ± 1.5 (0–8)	-	-
SOCIAL DOMAIN				
Social support (SSL-I12)	336	19.6 ± 6.5 (0–36)	268	17.2 ± 6.6 (0–35)
Loneliness (the Loneliness Scale—deJong Gierveld)	341	2.3 ± 2.9 (0–11)	273	2.7 ± 3.1 (0–11)
Financial problems (EORTC QLQ-C30)	361	9.2 ± 21.1 (0–100)	273	8.4 ± 18.9 (0–100)
Problems with social contacts (EORTC HN35)	361	4.9 ± 10.2 (0–83.3)	272	5.3 ± 12.2 (0–73.3)
LIFESTYLE DOMAIN				
Daily smoking				
Yes	91	25.2%	54	15.0%
No	254	70.4%	225	62.3%
Missing	16	4.4%	82	22.7%
Excessive alcohol consumption				
Yes	91	25.2%	40	11.1%
No	257	71.2%	239	66.2%
Missing	13	3.6%	82	22.7%
BMI (Kg/m^2^)	293	25.6 ± 4.8 (15.5–44.3)	297	24.1 ± 4.3 (14.0–42.2)
Physical activity (PASE)	345	97.5 ± 80.2 (0–535)	277	115.8 ± 98.5 (0–627)
Stress (IES-R)				
Intrusion	328	4.4 ± 4.7 (0–24)	264	3.3 ± 4.4 (0–24)
Avoidance	330	3.3 ± 3.7 (0–24)	261	2.3 ± 3.4 (0–18)
Hyperarousal	328	2.1 ± 2.8 (0–19)	265	1.6 ± 2.6 (0–14)
Numbing	331	1.2 ±1.5 (0–7)	263	0.7 ±1.3 (0–7)
Sleep disturbances	330	1.6 ± 2.1 (0–12)	264	1.4 ± 2.1 (0–11)

^1^ Missing measurements at 6 months by study design. Abbreviations: NEO-FFI = NEO Five Factor Inventory. SWAL-QoL = Swallowing Quality of Life questionnaire. FROG = Functional Rehabilitation Outcomes Grade. MNA = Mini nutritional Assessment. WHO = World Health Organization. MFI = Multidimensional Fatigue Inventory. IADL = Instrumental Activities of Daily Living. CFQ = Cognitive Failure Questionnaire. HADS = Hospital Anxiety and Depression Scale. PSQI = Pittsburgh Sleep Quality Index. MAC = Mental Adjustment to Cancer. UCL = Utrecht Coping List. PSMS = Pearlin and Schooler’s Sense of Mastery Scale. GSE = General Self = Efficacy Scale. BMI = Body Mass Index. PASE = Physical Activity Scale for the Elderly. IES-R = Impact of Event Scale-Revised. SSL = Social support Social Support List—Interactions. EORTC HN35 = European Organization for Research and Treatment of Cancer Quality of Life Questionnaire Head and Neck cancer-specific module. EORTC QLQ-C30 = European Organization for Research and Treatment of Cancer Quality of Life Questionnaire C30.

**Table 2 cancers-15-01603-t002:** The change in social eating problems from baseline to 24 months after treatment: observed and estimated values.

EORTC QLQ HN35—Social Eating Scale											
	Observed Data						Estimated with LMM			*p*-Value	*p*-Value Compared to T0
Measurement	N	Mean	SD	Median	IQR		Mean	99% CI			
Baseline	361	10.8	15.6	0.0	0.0	16.7	10.8	8.11	13.4	<0.001	-
3 months	298	22.4	23.8	16.7	0.0	33.3	22.7	19.8	25.6	-	<0.001
6 months	270	16.0	19.3	8.3	0.0	25.0	16.5	13.5	19.6	-	<0.001
12 months	246	12.3	19.1	0.0	0.0	16.7	13.9	10.8	17.0	-	0.009
24 months	217	9.4	17.1	0.0	0.0	12.5	11.5	8.2	14.7	-	0.57

Abbreviations: SD = standard deviation. IQR = interquartile range. LMM = linear mixed model. T0 = baseline. CI = confidence interval. EORTC HN35 = European Organization for Research and Treatment of Cancer Quality of Life Questionnaire Head and Neck cancer-specific module.

**Table 3 cancers-15-01603-t003:** Associations between social eating problems (EORTC H&N35) up to 24-month follow-up and baseline and 6 months post-treatment variables.

	Baseline			6 Months		
	F (*p*-Value)			F (*p*-Value)		
	Univariable	Multivariable Within-Domain	Multivariable Across-Domains	Univariable	Multivariable Within-Domain	Multivariable Across-Domains
HNC-SPECIFIC DOMAIN						
Swallowing-related Quality of Life (SWAL-QoL Total)	**7.257 (<0.001)**	**1.057 (<0.001)**	**9.906 (<0.001)**	3.195 (0.042)	n.s.	-
Perceived swallowing symptoms (SWAL-QoL Symptoms)	1.564 (0.182)	**3.530 (0.007)**	**4.173 (0.002)**	1.309 (0.271)	n.s.	-
Oral Function (FROG)	2.009 (0.091)	n.s.	-	1.276 (0.280)	n.s.	-
Nutritional status (MNA)	2.634 (0.033)	3.153 (0.014)	**4.692 (0.001)**	3.180 (0.043)	3.180 (0.043)	**6.089 (0.002)**
CLINICAL DOMAIN						
Tumor site	**2.454 (0.004)**	**2.454 (0.004)**	**2.724 (0.001)**	1.794 (0.099)	n.s.	-
Tumor stage	2.074 (0.016)	n.s.	-	0.645 (0.694)	n.s.	-
Chemotherapy	3.171 (0.013)	n.s.	-	3.570 (0.029)	3.570 (0.029)	n.s.
HPV status ^1^	0.502 (0.734)	-	-	0.099 (0.905)	-	-
Comorbidity	1.054 (0.378)	n.s.	-	1.109 (0.331)	n.s.	-
Performance status (WHO)	2.243 (0.063)	n.s.	-	0.182 (0.833)	n.s.	-
Lifetime depression diagnosis	1.017 (0.397)	n.s.	-	1.253 (0.287)	n.s.	-
PERSONAL DOMAIN						
Age	**3.725 (0.005)**	**3.753 (0.005)**	**3.627 (0.006)**	**5.056 (0.007)**	**5.129 (0.006)**	**5.727 (0.004)**
Sex	0.560 (0.692)	n.s.	-	1.596 (0.204)	n.s.	-
Education level	1.914 (0.055)	2.200 (0.025)	n.s.	1.144 (0.336)	n.s.	-
Living arrangements	1.000 (0.406)	n.s.	-	1.036 (0.356)	n.s.	-
Personality (NEO-FFI)						
Neuroticism	3.225 (0.012)	**4.787 (0.001)**	n.s.	2.064 (0.128)	n.s.	-
Extraversion	1.757 (0.135)	n.s.	-	0.485 (0.616)	n.s.	-
Openness to experience	1.676 (0.153)	n.s.	-	1.792 (0.168)	n.s.	-
Agreeableness	1.365 (0.244)	n.s.	-	4.340 (0.014)	4.450 (0.012)	n.s.
Conscientiousness	0.533 (0.711)	n.s.	-	1.046 (0.352)	n.s.	-
PHYSICAL DOMAIN						
Muscle strength (hand grip test)	1.751 (0.137)	n.s.	-	3.341 (0.036)	3.866 (0.022)	**5.218 (0.006)**
Hearing (Caron questionnaire)	0.864 (0.485)	n.s.	-	3.566 (0.029)	**4.708 (0.010)**	**5.155 (0.006)**
Physical fatigue (MFI)	0.508 (0.730)	n.s.	-	3.235 (0.040)	4.064 (0.018)	n.s.
Independence in daily activities (IADL)	0.908 (0.458)	n.s.	-	0.072 (0.930)	n.s.	-
Peak Expiratory Flow	1.658 (0.158)	n.s.	-	1.654 (0.193)	n.s.	-
PSYCHOLOGICAL DOMAIN—RESOURCES						
Positive adjustment to cancer (MAC)	2.865 (0.022)	n.s.	-	3.400 (0.034)	-	n.s.
Coping (UCL) ^2^						
Active coping	3.195 (0.013)	3.195 (0.013)	n.s.	-	-	-
Palliative coping	1.065 (0.372)	n.s.	-	-	-	-
Seeking support	1.220 (0.300)	n.s.	-	-	-	-
Reassuring thoughts	0.565 (0.688)	n.s.	-	-	-	-
Personal control/Mastery (PSMS) ^2^	1.544 (0.187)	n.s.	-	-	-	-
General Self-efficacy (GSE) ^2^	2.756 (0.027)	n.s.	-	-	-	-
PSYCHOLOGICAL DOMAIN—PROBLEMS						
Cognitive failure (CFQ)	1.435 (0.220)	n.s.	-	3.499 (0.031)	n.s.	-
Mental fatigue (MFI)	2.697 (0.030)	n.s.	-	3.259 (0.039)	n.s.	-
Anxiety symptoms (HADS)	**7.406 (<0.001)**	n.s.	-	**4.977 (0.007)**	n.s.	-
Depressive symptoms (HADS)	**6.424 (<0.001)**	**6.424 (<0.001)**	**5.914 (<0.001)**	**6.728 (0.001)**	n.s.	-
Fear of Cancer Recurrence (CWS)	**6.229 (<0.001)**	n.s.	-	**6.056 (0.003)**	**6.056 (0.003)**	n.s.
Negative adjustment to cancer (MAC)	2.790 (0.025)	n.s.	-	2.455 (0.087)	n.s.	-
Sleep problems (PSQI)	1.067 (0.372)	n.s.	-	4.121 (0.017)	n.s.	-
Coping (UCL) ^2^						
Avoidance coping	0.120 (0.975)	n.s.	-	-	-	-
Passive coping	**4.573 (0.001)**	n.s.	-	-	-	-
Expression of negative emotions	**4.571 (0.001)**	n.s.	-	-	-	-
SOCIAL DOMAIN						
Social support (SSL-I12)	1.044 (0.383)	n.s.	-	0.232 (0.793)	n.s.	-
Loneliness (the Loneliness Scale)	0.902 (0.462)	n.s.	-	0.754 (0.471)	n.s.	-
Financial problems (EORTC QLQ-C30)	2.832 (0.024)	2.875 (0.022)	n.s.	0.321 (0.726)	n.s.	-
Problems with social contacts (EORTC HN35)	2.448 (0.045)	2.388 (0.049)	n.s.	0.305 (0.738)	n.s.	-
LIFESTYLE DOMAIN						
Daily smoking	0.423 (0.792)	n.s.	-	1.662 (0.191)	n.s.	-
Excessive alcohol consumption	1.482 (0.206)	n.s.	-	2.904 (0.056)	n.s.	-
BMI	1.265 (0.282)	n.s.	-	1.015 (0.363)	n.s.	-
Physical activity (PASE)	0.726 (0.575)	n.s.	-	1.199 (0.303)	n.s.	-
Stress (IES-R)						
Intrusion	2.881 (0.022)	2.881 (0.022)	n.s.	**7.222 (0.001)**	n.s.	-
Avoidance	2.482 (0.042)	n.s.	-	2.576 (0.077)	n.s.	-
Hyperarousal	**3.585 (0.007)**	n.s.	-	**7.539 (0.001)**	n.s.	-
Numbing	1.774 (0.132)	n.s.	-	2.303 (0.101)	n.s.	-
Sleep disturbances	**3.562 (0.007)**	n.s.	-	**7.247 (0.001)**	**7.247 (0.001)**	n.s.

*p*-values < 0.01 are highlighted in bold. n.s. = not significant (*p* < 0.05), after the backward selection procedure. ^1^ HPV was only available in a subset of the NET-QUBIC patients (N = 199 (55.1%), of which 166 (83.4%) had oropharyngeal cancer) and was far from statistical significance (*p* < 0.05) in the univariable models, and thus not investigated in the multivariable models. ^2^ Missing measurements at 6 months by study design. Abbreviations: NEO-FFI = NEO Five Factor Inventory. SWAL-QoL = Swallowing Quality of Life questionnaire. FROG = Functional Rehabilitation Outcomes Grade. MNA = Mini nutritional Assessment. WHO = World Health Organization. MFI = Multidimensional Fatigue Inventory. IADL = Instrumental Activities of Daily Living. CFQ = Cognitive Failure Questionnaire. HADS = Hospital Anxiety and Depression Scale. PSQI = Pittsburgh Sleep Quality Index. MAC = Mental Adjustment to Cancer. UCL = Utrecht Coping List. PSMS = Pearlin and Schooler’s Sense of Mastery Scale. GSE = General Self = Efficacy Scale. BMI = Body Mass Index. PASE = Physical Activity Scale for the Elderly. IES-R = Impact of Event Scale-Revised. SSL = Social support Social Support List—Interactions. EORTC HN35 = European Organization for Research and Treatment of Cancer Quality of Life Questionnaire Head and Neck cancer-specific module. EORTC QLQ-C30 = European Organization for Research and Treatment of Cancer Quality of Life Questionnaire C3.

## Data Availability

The collection and integration of large amounts of personal, biological, genetic, and diagnostic information precludes open access to the NET-QUBIC research data. In the section, data and sample dissemination (www.kubusproject.nl) describes how the data are made available for the research community.

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
