# Peer review of "The Change in Social Eating over Time in People with Head and Neck Cancer Treated with Primary (Chemo)Radiotherapy: The Role of Swallowing, Oral Function, and Nutritional Status"

_cancers, 2023, doi:10.3390/cancers15051603_

Round 1

Reviewer 1 Report

This paper reports on the longitudinal assessment of social eating in patients with head and neck cancer treated with (chemo)radiotherapy. This is an important topic that is approached in a comprehensive and well planned manner in a large cohort of individuals.

The paper is well written and well presented, in particular the tables and graphs are very clear.

My main recommendations relate to clarification of terminology and simplification of language.

I was really unsure what the course of social eating is and remained unconvinced that this was the best term to describe what I would understand as change in social eating over time. Consider change in terminology and perhaps some more clarification of what you mean. A more extensive description of what social eating is would also be helpful.

Please avoid labelling patients with their disease – so people with Head and neck cancer and not head and neck cancer.

When listing association of social eating difficulties with a very long list of variables (summary and discussion) please group these variables according to categories a little more – currently they are a long lis with a bit of a random order of demographics variables, clinical variables, cancer outcomes in no particular order. Highlighting their categories would make a reader remember better.

When you describe association, indicate direction of association – for example if the course of social eating problem were associated with swallowing – does that mean that worsening or social eating problems was associated with better swallowing or worse swallowing? Etc etc

Be careful of double negative Is decrease in social eating problems an improvement or worsening of social eating

Please note I do not provide specific sentences that require clarification as this pattern is observed in multiple locations across the manuscript

I would suggest reframing your implications from “time matters” to “regular monitoring over time matters”

Keep the language simple - not “scant numerousness” but “small numbers”

I do not understand the statement ïn the discussion “study supports the multifaceted course of problems” – do you mean  - this study provides evidence of multifaceted nature of social eating difficulties over time??

Please include a brief recommendation/Implication in the conclusions. This is an important study and the current conclusions are a little underwhelming and do not really do its justice.

Author Response

Thank you very much for your review of our manuscript and the valuable comments and suggestions. We have provided our responses (shown in red) to your comments below and revised the manuscript accordingly.

This paper reports on the longitudinal assessment of social eating in patients with head and neck cancer treated with (chemo)radiotherapy. This is an important topic that is approached in a comprehensive and well planned manner in a large cohort of individuals.

The paper is well written and well presented, in particular the tables and graphs are very clear.

My main recommendations relate to clarification of terminology and simplification of language.

I was really unsure what the course of social eating is and remained unconvinced that this was the best term to describe what I would understand as change in social eating over time. Consider change in terminology and perhaps some more clarification of what you mean. A more extensive description of what social eating is would also be helpful.

We thank the reviewer for highlighting the need to clarify the main topic of this work. We agree that the phrase “change over time” better expresses the construct under study and used it in place of the word “course” throughout the manuscript. Also, an extended description of social eating was provided to readers (lines 70-74).

Please avoid labelling patients with their disease – so people with Head and neck cancer and not head and neck cancer.

We thank the Reviewer for highlighting the need to improve the language use of this manuscript. We changed labelling phrases (such as, “HNC patients”) with descriptive expressions (e.g., “people with HNC”) throughout the manuscript.

When listing association of social eating difficulties with a very long list of variables (summary and discussion) please group these variables according to categories a little more – currently they are a long list with a bit of a random order of demographics variables, clinical variables, cancer outcomes in no particular order. Highlighting their categories would make a reader remember better.

Throughout summary and discussion sessions, we organized and linked the lists of variables associated with the change in social eating problems according to the following categories: HNC-specific, clinical, personal, physical, psychological, social, and lifestyle domains (lines 340-343, 418-421). Where it was not possible to name both variables and categories due to word-count constraints, variables were ordered according to categories and the name of categories was omitted (lines 37-38).

When you describe association, indicate direction of association – for example if the course of social eating problem were associated with swallowing – does that mean that worsening or social eating problems was associated with better swallowing or worse swallowing? Etc etc

Different patterns of association were found in our study, with the same variable associated in different directions at different time points with changes in social eating problems. In the Results and Discussion sections, we described these patterns with reference to individual variables (e.g., lines 283-297, 327-335). We chose to present and discuss our results in this fashion in order to minimize confusion for the reader and portray the most accurate picture possible. For these reasons, when associated variables were listed in the discussion, summary, and abstract sections, we did not provide the direction of the associations.

Be careful of double negative Is decrease in social eating problems an improvement or worsening of social eating.

Please note I do not provide specific sentences that require clarification as this pattern is observed in multiple locations across the manuscript.

We used the expression “social eating problems” throughout the manuscript to describe our outcome in the most possible adherent way with EORTC social eating questions. In fact, EORTC QLQ HN35 questions are phased in terms of problems, asking participants to rate how much trouble for them is eating in front of other people and enjoying their meals. Nevertheless, we recognize double negative may raise interpretative issues. Thus, we expanded the interpretation of the results throughout the manuscript (lines 283-297, 328-333, 370-376, 388-389).

I would suggest reframing your implications from “time matters” to “regular monitoring over time matters”

Suggested changes were made (line 357).

Keep the language simple - not “scant numerousness” but “small numbers”

The suggested change was made (line 404).

I do not understand the statement in the discussion “study supports the multifaceted course of problems” – do you mean  - this study provides evidence of multifaceted nature of social eating difficulties over time??

We thank the Reviewer for rephrasing this expression in such a clear way and we provided changes at lines 415-416.

Please include a brief recommendation/Implication in the conclusions. This is an important study and the current conclusions are a little underwhelming and do not really do its justice.

We expanded the Conclusion with recommendations for supportive care targeting social eating based on study results (lines 421-426).

Reviewer 2 Report

The Authors reported on a prospective cohort study (NETherlands QUality of life and BIomedical Cohort), which was aimed at investigating social eating problems and their course in HN cancer patients treated with definitive chemoradiation. The association between social eating and swallowing, oral function and nutritional status was tested. The topic is of interest, methodology is robust, results are clear, and the discussion well elaborated on.

My only suggestion would be to try to visually portray the most important correlations found to build causal models. It would be interesting to hypothesize some causal inference diagrams using DAG (directed acyclic graphs).

Author Response

Thank you very much for your review of our manuscript and the valuable comments and suggestions. We have provided our responses (shown in red) to your comments in this letter and revised the manuscript accordingly.

The Authors reported on a prospective cohort study (NETherlands QUality of life and BIomedical Cohort), which was aimed at investigating social eating problems and their course in HN cancer patients treated with definitive chemoradiation. The association between social eating and swallowing, oral function and nutritional status was tested. The topic is of interest, methodology is robust, results are clear, and the discussion well elaborated on.

My only suggestion would be to try to visually portray the most important correlations found to build causal models. It would be interesting to hypothesize some causal inference diagrams using DAG (directed acyclic graphs).

We thank the Reviewer for this input to enhance the visualization of our results and strengthen the discussion. We built DAGs which depict hypothesized causal relationships between the change in social eating problems over time and associated variables in the overall models (Figure 5). In the DAGs, the proximity of each variable to “change in social eating” is based on the F values of the overall models (closer variables presented with higher F values). We did not portray possible causal relationships among the investigated exposure variables, as this would have fallen beyond the scope of our study and would have needed a thorough revision of the literature and further analyses of collected data. The DAGs were presented in the Discussion section since they represent the Authors’ hypotheses generated based on the study results (lines 343-344).